# Nanohybrid Antifungals for Control of Plant Diseases: Current Status and Future Perspectives

**DOI:** 10.3390/jof7010048

**Published:** 2021-01-13

**Authors:** Mousa A. Alghuthaymi, Rajkuberan C., Rajiv P., Anu Kalia, Kanchan Bhardwaj, Prerna Bhardwaj, Kamel A. Abd-Elsalam, Martin Valis, Kamil Kuca

**Affiliations:** 1Biology Department, Science and Humanities College, Shaqra University, Alquwayiyah 11971, Saudi Arabia; mosa-4507@hotmail.com; 2Department of Biotechnology, Karpagam Academy of Higher Education, Coimbatore 641021, Tamil Nadu, India; kuberan87@gmail.com (R.C.); rajivsmart15@gmail.com (R.P.); 3Electron Microscopy and Nanoscience Laboratory, Department of Soil Science, College of Agriculture, Punjab Agricultural University, Ludhiana 141004, Punjab, India; 4School of Biological and Environmental Sciences, Shoolini University of Biotechnology and Management Sciences, Solan 173229, Himachal Pradesh, India; kanchankannu1992@gmail.com (K.B.); prernabhardwaj135@gmail.com (P.B.); 5Plant Pathology Research Institute, Agricultural Research Center (ARC), Giza 12619, Egypt; 6Department of Neurology of the Medical Faculty of Charles University and University Hospital in Hradec Kralove, Sokolska 581, 50005 Hradec Kralove, Czech Republic; martin.valis@fnhk.cz; 7Department of Chemistry, Faculty of Science, University of Hradec Kralove, 50003 Hradec Kralove, Czech Republic; 8Biomedical Research Center, University Hospital in Hradec Kralove, Sokolska 581, 50005 Hradec Kralove, Czech Republic

**Keywords:** chitosan, nanohybrids, polymer-metal composites, antifungal, postharvest

## Abstract

The changing climatic conditions have led to the concurrent emergence of virulent microbial pathogens that attack crop plants and exhibit yield and quality deterring impacts on the affected crop. To counteract, the widespread infections of fungal pathogens and post-harvest diseases it is highly warranted to develop sustainable techniques and tools bypassing traditional agriculture practices. Nanotechnology offers a solution to the problems in disease management in a simple lucid way. These technologies are revolutionizing the scientific/industrial sectors. Likewise, in agriculture, the nano-based tools are of great promise particularly for the development of potent formulations ensuring proper delivery of agrochemicals, nutrients, pesticides/insecticides, and even growth regulators for enhanced use efficiency. The development of novel nanocomposites for improved management of fungal diseases can mitigate the emergence of resilient and persistent fungal pathogens and the loss of crop produce due to diseases they cause. Therefore, in this review, we collectively manifest the role of nanocomposites for the management of fungal diseases.

## 1. Introduction

Agriculture is the major backbone of a country’s economy; in recent years, the agricultural production and yields have been increased along with pesticides and insecticides usage [1,2,3]. But in the present scenario, the practices of agriculture methods are fretful about the extensive usage of chemical pesticides used to combat microbes/insects. Among microbes, the fungal genera belonging to class Ascomycetes (*Verticillium*, *Alternaria*, and *Fusarium*) and Basidiomycetes (*Rhizoctonia*, *Sclerotium*) can cause growth as well as yield-deterring effects on plants leading to significant economic losses to the farmers [4]. To curb these disease manifestations, a stringent dynamic approach should be implemented with novel technology encompassing smart materials with biological ingredients for sustainable delivery with prolonged efficiency.

Recently, nanoscale engineering approaches have posed a new advanced entity derived from a biological source and self-assembling systems [5]. At present, the nano concept is playing an immense role in medicine and pharmacology; in both these fields’ nanotechnology has attained a decisive role in drug delivery, diagnosis, imaging, antimicrobial agents, and sensors [6]. In the agriculture sector, nanotechnology products and devices are being eventually utilized in plant hormone delivery, nano-barcoding, development of rapid and sensitive nano-sensor systems for easy diagnosis of diseases, pests, and nutritional deficiencies, the targeted/controlled/slow-release of agrochemicals, seed germination enhancers, nano-vectors for efficient gene transfer and several other applications [7].

Engineered nanoparticles (NPs) possess the desired size and shape with specific optical properties that enable them to be used for various agricultural applications particular instance is as novel pesticide formulations exhibiting improved pest and pathogen control efficiencies [8,9,10]. The most widely used nanoparticles for the control of plant diseases are carbon, silver, silica, and non-metal oxides or alumino-silicates. The research studies performed on carbon nanomaterials have shown diverse and promising agri-applications including the promotion of plant growth and development, besides effective control of several plant pathogens such as *Xanthomonas*, *Aspergillus* spp., *Botrytis cinerea*, and *Fusarium* spp. A study revealed that silica nanoparticles were effective in maize conferring resistance to the phytopathogens such as *Fusarium oxysporum* and *Aspergillus niger* [11]. However, the action spectrum and pest/pathogen control efficacies of the nano-enabled pesticides can be improved through the development of nano-hybrids or composites [12,13]. The components of the nano-hybrids or nanocomposites can have diverse chemical origins spanning over biological-inorganic, as well as natural/synthetic organic-inorganic materials. These composites do not involve physical mixing of the components and therefore, possess peculiar properties which may or may not essentially represent additive or augmenting effect considering the properties of the individual components [12]. Escalated research interest for the development of potent, effective and multi-functional anti-microbial nano-hybrids has been witnessed in the present decade for instance, alumino-silicate nanoplates have been used for the development of pesticide formulations that exhibit twin benefits of improved biological activity and better environmental safety compared to use of engineered NPs [14]. Therefore, nano-formulated particles/composites have the potential to tackle disease outbreaks caused by fungal pathogens effectively. In this review, we emphasize the development of antifungal nano-hybrids encompassing conjugates of organic or inorganic molecules, biological components, and biopolymers to develop the cheaper, reliable, and effective product(s) against most fungal pathogens of plants (Figure 1).

## 2. Nanotechnology Advances for Improved Pathogen Diagnosis

In agriculture, the outbreak of pathogenic infections is to be monitored at the earliest, else the crop yield might get heavily compromised. Therefore, prompt diagnostic methods are urgently required as the old conventional methods such as immunological techniques and other molecular tools require specialized skilled man-power and are not cost-effective. To pace up with the rapid spread of the plant pathogenic infections, rapid, robust, sensitive, and low-cost smart material based diagnostic protocols are required to be designed for counteracting bacterial and fungal infections in agriculture crops.

Nanotechnology-based pathogen diagnosis is gaining overwhelming attention from the research community due to the functional optical properties and ease of handling technology of these materials [15]. The added advantage of nanotechnology is that nanoparticles can be conjugated with nucleic acids, proteins, and other biomolecules, a feature that enables rapid, sensitive, and reliable diagnosis of pathogens [16]. Among the various nanomaterials, quantum dots are a special class of nanocrystals that exhibit tunable size-dependent fluorescence characteristics for which these are explored widely in agriculture and allied sectors. A specific quantum dot-based nano-sensor has been developed for diagnosing *Candidatus Phytoplasma aurantifolia* in the infected lime even at low occurrence of 5 phytoplasma cells µL^−1^ [17]. Fluorescent silica nanoparticles conjugated with antibody molecules can rapidly detect the *Xanthomonas axonopodis* pv. *vesicatoria* a causative agent for spot diseases in tomato and pepper [18].

Gold nanoparticles are widely used in pathogen diagnosis due to the unique optical or electrochemical properties enabling simple and easy protocols for the quick diagnosis of pathogens. Singh et al. [19] developed an immunosensor based on nanogold using Surface Plasmon Resonance (SPR) that could detect the Karnal bunt disease in wheat (*Tilletia indica*). Wang et al. [20] developed an electrochemical sensor comprised of copper nanoparticles to detect the fungus *Sclerotinia sclerotiorum* in oilseeds. They have utilized this electrochemical sensor to measure the level of salicyclic acid accurately. Schwenkbier et al. [21] developed a chip-based hybridization technique incorporating silver nanoparticles for the detection of *Phytophthora* species. Copper oxide nanoparticles and nanolayers have been synthesized and applied for easy detection of *Aspergillus niger* in crop plants. In another approach, portable equipment was used to detect bacterial, fungal species in stored food grains. Likewise, Ariffin et al. [22] formulated a nanowire biosensor to detect Cauliflower Mosaic Virus and Papaya Ring Spot Virus. Thus, with the above scientific evidence, it can be identified that nano-based sensors and kits play a vital role in crop health care including products for rapid testing, disease diagnostics, and environmental monitoring aspects.

## 3. Nanocomposites and Their Mode of Action on the Fungal Phytopathogens

Nanocomposite materials include multi-phase components. These materials may be comprised of components with variable phase domains with atleast one continuous phase while another having nano-scale dimensions [23]. These hybrid nanomaterials can be generated through co-synthesis/impregnation of diverse inorganic and organic components [24]. Nanocomposites have been extensively studied due to the properties of inorganic and organic materials that enact concurrently to perform the desired activity [25]. Generally, nanocomposites are derived by the addition of nano-particulate materials in long-chain or short-chain polymeric matrices. The derived nanocomposites exhibit improved properties not observed for any of the individual components. Most likely, the combination of polymers with nanoparticles is anticipated to increase the properties of the polymer significantly [26]. Such kind of nanocomposites are now widely being used in food processing, pest detection and management, food health screening, water treatment, disease detection, drug-delivery systems, and improvement of sustainable agriculture [27,28,29,30]. Likewise, the polymer composites act as fertilizers which increase the nutrients uptake, decrease soil toxicity [31,32]. Moreover, nanocomposites are well being used to increase the shelf life of food materials by acting as antimicrobial dispositions and as sensors [33].

Plant disease management using hybrid polymer nanocomposites is focused on making mulch films to control weeds; as nano pesticides and as a biostatic agent [34]. Min et al. [35] demonstrated that silver nanoparticles can effectively inhibit the phytopathogens such as *Rhizoctonia solani*, *Sclerotinia sclerotiorum*, and *S. minor*. Further, silver nanoparticles can cause extensive damage by breaking the hyphal wall membrane followed by internal damage of the hyphae. Sepiolite, a magnesium silicate, was blended with MgO to form a (SE-MgO) nanocomposite that exhibited excellent antifungal activity against rice pathogen *Fusarium verticillioides*, *Bipolaris oryzae*, and *Fusarium fujikuroi* with ED90 > 249 μg/mL compared with MgONPs [36].

Carbon nanomaterials have shown to possess strong antifungal activity against pathogens *Fusarium graminearum* and *Fusarium poae* [37]. Six carbon nanomaterials (CNMs), single-walled carbon nanotubes (SWCNTs), multi-walled carbon nanotubes (MWCNTs), graphene oxide (GO), reduced graphene oxide (rGO), fullerene (C_60_), and activated carbon (AC) were evaluated against the phytopathogens. The outcome imposes that SWCNTs, MWCNTs, GO and rGO at varying concentrations decreased the biomass, mycelia growth and inhibited spore germination of the fungal pathogens. However, the C_60_ and AC didn’t exert any activity against the tested pathogens.

In another study, GO-AgNPs nanocomposite was synthesized through interfacial self-assembly and was evaluated for its antifungal activity against the pathogen *Fusarium graminearum* under in vitro and in vivo conditions [38]. The fungicidal activity of GO-AgNPs against the *F. graminearum* spores was 4.68 µg/mL; the nanocomposite also constrained the germination of spores and hyphae. Moreover, the authors demonstrated response to GO-AgNPs through the SEM analysis of the events of spore germination of *F. graminearum*. The images portray that *F. graminearum* conidia were crumpled, widened, and damaged heavily as seen in Figure 2. The fungicidal activity of nanocomposite is due to the adsorption of nanoparticles to the fungal cell wall membranes and oxygen groups of GO form hydrogen bond with lipopolysaccharide subunits of the cell membrane which contains sugars, phosphates, and lipids.

Downy mildew is a disease caused by *Plasmopara viticola* in grapes plants that leads to extremely heavy yield loss. To combat the disease, a nanocomposite construct consisting of Graphene oxide (GO) and Iron oxide (Fe_3_O_4_) as GO-Fe_3_O_4_ nanocomposite was developed [39]. The study demonstrated that the pretreatment of the leaf discs with nanocomposite followed by inoculation with *P. viticola* sporangium suspension strongly inhibited the spore germination. This effect may be attributed to the blockage of the water channels of sporangia by surface adsorption of the nanocomposites. From the images, it is evident that bare GO and Fe_3_O_4_ possessed moderate spore germination inhibition activity while the nanocomposites triggered stupendous activity [39].

Silver-Titanate nanotubes (AgTNTs) nanocomposite was synthesized through a one-pot chemical method and functionalization with AgNPs. These nanocomposites were further evaluated against the phytopathogenic fungi *Botrytis cinerea* by the photoinactivation method. The nanocomposite stimulated Reactive Oxygen Species (ROS) cascades and damaged the conidia which eventually led to cell death [40]. In a microwave-assisted method, a magnetically separable Fe_3_O_4_/ZnO/AgBr nanocomposite was synthesized. These synthesized nanocomposites inactivated the *Fusarium graminearum* and *Fusarium oxysporum* within a short period of 120 and 60 min. Thus, the efficacy of nanocomposites can be identified as the combined aggregation of the inorganic metal complexes [41].

The leaf extract of *Adhatoda vasica* was utilized as a reducing agent to synthesize Copper oxide nanoparticles/Carbon nanocomposite through the green chemistry approach [42]. These nanocomposites exhibited effective growth inhibitory activity against *Aspergillus niger*. Thus, from these reports, it is evident that inorganic metals possessing inherent properties when combined as nanocomposites as a result of synergistic effects result in the fabrication of nanocomposites which exhibit activities on comparison with bare metals. However, the behavior of metal alloys during nanocomposite synthesis will differ according to the synthesis methods; elemental composition, and the applications.

Nanocomposites can be manufactured from any combination of materials like polymers, metals, and ceramics [43]. Among the materials, polymers and inorganic/organic materials will have a high aspect ratio and surface properties which enable them to be widely used in a different range of industries. In agriculture, polymers play an indispensable role in the release of chemical moieties such as fungicides, insecticides, growth stimulants and germicides [44]. The promising advantage of polymers is to control the release rate and rate of biodegradability of the embedded or encapsulated compound. These features make polymers widely used as a delivery agent in medicine and agrochemicals [45,46]. Polymers such as cellulose acetate phthalate, gelatin, chitosan, gum Arabic, polylactic acid, poly-butadiene, poly-lactic-glycolic acid, polyhydroxyalkanoates, polyvinyl alcohol (PVA), polyacrylamide, and polystyrene are widely used as delivery agents for drugs and agrochemicals [13,23].

In agriculture, the usage of polymer nanocomposites is to be critically chosen according to the application. Further, natural polymers are preferably chosen for the agri-applications owing to the nature of degradability and controlled release behavior. Chitosan polymer is gaining a new avenue in the plant protection field due to its outstanding properties. Nanochitosan exhibits anti-microbial potentials against bacteria and fungi at varying levels of concentrations. Combined formulation comprised of metal or metal oxide nanomaterials encapsulated or embedded in chitosan exhibit improved antimicrobial potentials. For instance, nanocomposite Ag-chitosan prominently exhibited antibacterial activity to a higher extent. Likewise, the fungicidal activity of clay chitosan nanocomposite was evaluated against *Penicillium digitatum* under in vivo and in vitro conditions [47]. The results were promising in terms of the superior activity exhibited by the nanocomposite formulation. Likewise, copper nanoparticles (Cu NPs), zinc oxide nanoparticles (ZnNPs), and chitosan, zinc oxide, and copper nanocomposites (CS–Zn-Cu NCs) were chemically fabricated and evaluated against plant pathogenic fungi *A. alternata*, *R. solani*, and *B. cinerea* [48]. The results intrinsically acclaimed that nanocomposite displayed higher activity at a concentration of 90 µg mL^−1^. Antifungal activities of Bimetallic blends and Zn-Chitosan, and Cu-Chitosan at concentrations of 30, 60, 90 µg mL^−1^ effectively inhibited the growth of *Rhizoctonia solani*. Further, it indicated effective control of cotton seedling damping-off under greenhouse conditions [49]. Silver/chitosan nanocomposite portrayed incremental growth-inhibitory effect against phytopathogens isolated from chickpea seeds [50]. Moreover, the individual metal and polymer components exhibited inhibitory activity lower than the Silver/Chitosan nanocomposite against the test pathogen, *Aspergillus niger*. A combination of chitosan/silica nanocomposite was evaluated against *Botrytis cinerea* under in vitro and in vivo (natural and artificial infections) conditions [51]. The in vitro study revealed complete reduction of the fungal growth by the nanocomposite compared to 72% and 76% inhibition potential of chitosan and silica nanoparticles respectively. Moreover, under natural conditions, the Chitosan/silica nanocomposite effectively hampered the gray mold disease in Italian grapes by 59% and in Benitaka grapes by 83% without affecting the grape quality.

The chitosan conjugated Ag nanoparticles functionalized with 4(*E*)-2-(3-hydroxynaphthalene-2-yl) diazenyl-1-benzoic acid were prepared which demonstrated improved effectiveness against *A. flavus* and *A. niger* forming larger inhibition zone of 20.2 mm and 27.0 mm respectively [52]. Likewise, chitosan hydrogel with cinnamic acid encapsulating *Mentha piperita* essential oil markedly inhibits the growth of mycelia of *A. flavus* at a concentration of 800 mg/mL [53]. Thus, with the above appropriate scientific evidence, it can be inferred that organic/inorganic-metal-polymer hybrid nanocomposites exhibit exceptionally superior anti-fungal activities under in vitro and in vivo conditions.

### 3.1. Antifungal Mechanism of Nanoparticles/Nanocomposites

A promising nano-fungicide should possess an equivalent or superior activity corresponding to the bulk metal at relatively lower concentrations. Moreover, it is desirable to understand the phyto and eco-toxicity issues due to the release of metal ions. Multifarious mechanisms were involved in the antifungal activity executed by nanomaterials. The generalized antifungal activity is provided in the Figure 3. The antifungal activity of nanomaterials can be accomplished by the following events. Generally, fungal cell wall and cell membrane architecture involves chitin, lipids, phospholipids and polysaccharides with specific predominance of mannoproteins, β-1,3-d-glucan and β-1,6-d-glucan proteins [54]. Internalization of the nanomaterials occurs through three mechanisms; (i) direct internalization of nanoparticles in the cell wall, (ii) specific receptor-mediated adsorption followed by internalization, (iii) internalization of nanomaterials through ion transport proteins [55]. Post-internalization, the nanomaterials may inhibit the enzyme β-glucan synthase thereby affecting the N-acetylglucosamine [N-acetyl-d-glucose-2-amine] synthesis in the cell wall of fungi. As a consequence of enzyme inhibition, abnormalities like enhanced thickening of the cell wall, liquefaction of cell membrane, dissolution or disorganization of the cytoplasmic organelles, hyper-vacuolization, and detachment of cell wall from cytoplasmic contents indicating incipient plasmolysis might occur [56].

At the molecular level, the nanomaterials interact with various biomolecules and form complexes with different biomolecules thereby causing structural deformation in the biomolecules, inactivation of the catalytic proteins, and nucleic acid abnormalities like DNA breakage, and chromosomal aberrations [57,58]. Reactive Oxygen Species (ROS) play a critical role in antifungal activity mechanism of nano-composite materials. The metal ions trigger ROS and damage the biomolecules leading to cell death. Further, to authenticate the role of ROS in antifungal mechanism; Lipovsky et al. [59] deciphered that the increased expression of lipid peroxidation is a clear indicator of ROS generation. Meanwhile, stress enzymes like superoxide dismutase, glutathione dismutase, ascorbate peroxidase were upregulated/downregulated upon nanomaterials treatment in fungi [60].

### 3.2. Nano-Hybrid Antifungals for Control of Toxigenic Fungi and Mycotoxins Degradations

Mycotoxins are the natural contaminants in food and feed worldwide. Changing climatic patterns have significantly affected the agricultural production due to limiting water, and land resources, temperature extremes and elevated humidity conditions [61]. The elevated humidity and temperature allowed for the proficient growth of a variety of mycotoxigenic fungal genera, *Aspergillus* spp., *Penicillium* spp., and *Fusarium* spp. which produce a variety of mycotoxins including aflatoxins (AFs), fumonisins (FBs), ochratoxins (OTs), trichothecenes (TCs), and zearalenone (ZEA). These mycotoxins cause detrimental health impacts in humans manifested as liver cancer, aflatoxicosis, malabsorption syndrome and reduction in bone strength [62].

Conventionally, mycotoxins can be detected by chromatographic techniques like High-Pressure Liquid Chromatography (HPLC), Gas chromatography-Mass Spectrometry (GC-MS), and Liquid Chromatography-Mass Spectrometry (LC-MS) [62,63]. These techniques are robust, sensitive and specific but have high cost of analysis per sample and are therefore, expensive and time consuming. Further, to advance with innovation a simple, cheap and sensitive technique can be achieved with the aid of nanotechnology. Nano-based mycotoxin detection, and management involves specific properties such as selectivity, sensitivity, simplicity and multiple capabilities [64]. Hybrid nanomaterials are a new paradigm to counteract mycotoxin management. Generically, hybrid nanomaterials are having superior properties and multimodality (simultaneous detection, detoxification, and management abilities) when compared with polymers/metals, organic molecules when used individually in mycotoxicology [65].

Hybrid nanomaterials consisting of polymers/metals/organic molecules can synergistically interact with each other and accelerate the reaction kinetics (Figure 4). In mycotoxin management, nanohybrid materials are used for detection, detoxification and management [61]. For instance, Bhardwaj et al. [66] developed an immunosensor comprising graphene quantum dots (GOD), gold nanoparticles (AuNPs). Further, GOD-AuNPs were fabricated onto an indium tin oxide (ITO) electrode modified with an antibody (anti-AFB1) (anti-AFB1/GQDs-AuNPs/ITO) to detect Aflatoxin (AFB1). The hybrid immunosensor detected the AFB1 with high sensitivity for the presence of aflatoxin B1 even at very low concentrations (0.1 to 3.0 ng/mL) in the food sample.

Another important concern is mycotoxin detoxification. Hybrid nanomaterials are smart detoxifying agents. The hybrid nanomaterials can be incorporated in feed to sequester mycotoxin by forming a complex in the gastrointestinal tract so that the severity of the toxin gets ceased. To detoxify mycotoxin, Hamza et al. [67] devised a hybrid nanomaterial comprised of β-glucan mannan lipid particles (GMLPs) encapsulating the humic acid nanoparticles (HA-FeNPs). The specificity of this hybrid material was that the β-glucan molecules produced 3 to 4 µm hollow porous microspheres. Moreover, the addition of humic acid increased the binding affinity of Aflatoxin B1. The bare GMLPs and HA showed a moderate binding affinity for aflatoxin (10.8 μg AFB1/mg HA for GMLP HA). However, the addition of Fe increased the adsorbent capacity for GMLP HA-FeNPs AFB1 mass to 13.5 μg AFB1/mg HA and 16.8 μg AFB1/mg. This study showcased that this nanohybrid material can be used as a safe detoxification agent.

Recent findings related to nanocomposites (NCPs) based on organic polymeric and inorganic matrices or hybrid materials as effective antifungal agents against mycotoxigenic fungi and mycotoxin reduction have been summarized by Jampílek and Kráĺová, [68]. Spadola et al. [13] have identified an interesting alternative technique to inhibit aflatoxin production in *Aspergillus flavus.* They have formulated nanoparticles of poly-ε-caprolactone polymer and loaded the generated nanoparticles with two thiosemicarbazone (benzophenone or valerophenone) compounds to curb mycotoxin production in *A. flavus.* Pirouz et al. [69] investigated the use of hybrid magnetic graphene oxides (MGOs) as an adsorbent for DON, ZEA, HT-2, and T-2 in naturally contaminated palm kernel cakes (PKC). At optimum reduction conditions at pH 6.2 for 5.2 h at 40.6 °C, the MGO was able to reduce the amount of DON, ZEA, HT-2. In the same direction, MGO adsorbents have been used to detoxify polluted AFB1 oils, their absorbents are made of MGO and magnetic reduced graphene oxides (MrGO) both of which are incorporated with Fe_3_O_4_ nanoparticles. The MGO and MrGO were renewable, however, after seven cycles, with no major losses in the adsorption activities [70]. Copper-chitosan nanocomposite-based chitosan hydrogels (Cu-Chit/NCs hydrogel) have been prepared using metal vapor synthesis (MVS). Also, SEM measurements revealed damage to *A. flavus* cell membranes. Current findings indicate that the antifungal activity of nanocomposites in vitro can be beneficial depending on the type of fungal strain and the concentration of nanocomposites (Figure 5A). Cu-Chit/NCS hydrogel is a revolutionary nanobiopesticide developed by MVS used in food and feed to induce plant protection against mycotoxigenic fungi [71]. The fungicidal behavior of chitosan-silver nanocomposites (Ag-Chit-NCs) against *Penicillium expansum* from the feed samples was investigated. Ag-Chit-NCs < 10 nm in size have an important antifungal inhibitory effect against *P. expansum*, the causative agent of blue mold-contaminated dairy cattle feed [72]. *P. expansum* treated with Ag-Chit *P. expansum* treated with Ag-Chit NCs was investigated by HR-SEM, alterations in conidiophores, metulae, phialides, and mature conidia characteristics had been observed to obtain information about the mode of action of Ag-Chit-NCs (Figure 5B). Therefore, nanocomposites can be utilized as viable alternatives to the already available arsenal of fungicides (Table 1).

To prevent fungal growth and mycotoxin production in food materials and packaging; hybrid nanofiber mats composed of cellulose acetate encapsulated with AgNPs were prepared by electrospinning. Due to close tight assembly of packaging in food materials, these nanofiber mats allowed low penetration of air permeability preventing fungal growth, and silver nanoparticles inherently inhibited the growth of yeast and molds [73]. Thus, these nanocomposite materials have shown great potential for future applications in the food packaging and preservation industry.

### 3.3. Postharvest Management of Nanocomposite Against Pathogenic Fungi

Perishable vegetables and fruits are spoiled due to transport, storage and growth of spoilage and opportunistic microbes. Therefore, microbial decay of fruits and vegetables is a great concern for researchers to formulate a driven strategy control measures with long-standing efficiency. Nanotechnology is an alternative solution to develop sustainable horticulture in preserving and managing post-harvest diseases of fruits and vegetables [74,75]. This technology offers various products such as packaging thin films; helping for labeling fresh products using the multiple chips (nanobiosensors), improvement of packaging appearance and prevention of the impact of gases and unsafe rays.

Conventionally, fungicides like imazalil, thiabendazole, pyrimethanil, fludioxonil and chloride-based chemicals have been used for management of post-harvest diseases of horticultural produce [61]. Though effective but prolonged use of these fungicides has led to development of among the fungal genera. Further, the active ingredients of the fungicides are toxic to humans and also to the ecosystem. Post-harvest diseases can be classified into two groups (i) diseases from field infection (ii) diseases due to post-harvest infection ([76,77]).

Citrus orange fruit, *Citrus sinensis* L. Osb., is often spoiled by *P. digitatum* during the post-harvest storage/transportation periods. To counteract against the pathogen a nanocomposite clay-chitosan nanocomposite (CCNC) was synthesized and evaluated under in-vitro conditions; at 20 µg/mL the nanocomposite completely inhibited the growth of *P. digitatum*. In *in-vivo* trials, the nanocomposite reduced the lesions by 70% and inhibited the disease in orange [47]. The CNCC coated orange were observed to be free from the disease, and exhibited high pH, chroma, peel moisture, and firmness in comparison with the control [47].

A chitosan-Titanium dioxide composite film (70 µm thickness) was used as a packaging material to extend shelf life during the postharvest storage of grapes by preventing spoilage microbial infection. The composite film enhanced the shelf life and resisted the mildew infection in stored grapes for up to 22 days [78]. Similarly, a nanocomposite of silver/gelatin/chitosan was applied as a hybrid film in grapes to improve the storage shelf-life under cold conditions. The hybrid film stored grapes didn’t show signs of infection, had a fresh appearance and showed no leakage of the grapes [79].

Banana (*Musa acuminata* L.) is a famed fruit consumed by all peoples and is considered to be one of the high-valued fruit in horticulture. During post-harvest, the banana fruits get deteriorated on storage and transportation periods due to its climacteric nature i.e., increased respiration rate. To overcome this and ensure delayed ripening, maintenance of fruit firmness and reduced mass loss of fruits coating of a nanocomposite containing soybean protein isolate, cinnamaldehyde, and ZnO NPs on the banana fruit was observed to be very effective [80]. In another study, a predominant disease (anthracnose) caused by *Colletotrichum musae* in banana causing a major loss to the farmers was prevented by use of metallic nanoparticles (silver, nickel, copper and magnesium) prepared from ajwain and neem leaf extract. In postharvest period, silver nanoparticles were sprayed on the banana at different concentrations ranging from 0.02 to 0.2 percent resulting in reduction in the anthracnose infection (6.67 percent disease index) on use of the least concentration of 0.2% AgNPs [81].

Apple (*Malus domestica* Bork) is a perishable fruit consumed by all ages worldwide. Since it is a climacteric fruit, it is indeed useful to pro-act for development of a sustainable methodology to prevent post-harvest losses of apple fruit. Polylactic acid incorporated with ZnO nanoparticles was applied as a thin film in a fresh-cut apple stored at 4 °C for 14 days. PLA-ZnO NPs exhibited effective inhibition of yeast and molds in fresh-cut apples. This showed the possibility that PLA nanocomposite can be used as a packaging material in apples during the storage period [82].

Mango (*Mangifera Indica* L.) is also supposed to be highly prone to infections by post-harvest pathogen(s) causing anthracnose diseases. A chitosan-silver NPs composite was prepared and evaluated for the antifungal effect against conidial germination of *C. gloeosporioides*. The prepared chitosan-silver nanocomposite (at 100 µg mL^−1^) completely suppressed the spore germination. Under in vivo conditions, mango fruits were inoculated with the fungal spores. The infected mango fruit were then coated with the aforementioned nanocomposite and evaluated for the disease incidence. The nanocomposite prominently lowered the disease incidence with 45.7% and 71.3% at 0.5 and 1.0% of nanocomposite respectively [83]. Likewise, a nanocomposite containing aloe vera gel, ZnOPs and glycerol was coated on an edible mango and stored for 9 days at room temperature. After the storage days, the edible mango didn’t show any sign of infection/diseases [84]. Similarly, nanoemulsion containing chitosan was effectively inhibited *C. musae* and *C. gloeosporioides*. However, the chitosan nanoemulsion showed better prospective results than chitosan nanocomposite in banana, papaya and dragon fruits [85]. New biopolymers composite oligochitosan (OCS) and OCS/nanosilica (OCS/nSiO_2_) hybrid materials with impressive synergistic action are likely to be considered potential protections for plants infected with *Colletotrichum* sp. Conjugated nanomaterials may be considered possible biotic elicitors that not only avoid anthracnose disease but also efficiently enhance plant growth [86]. The same team found that silica and hybrid material had good antifungal properties against *P. infestans*, the causal organism for late blight in tomato and potato, but the antifungal properties of hybrid materials, due to their synergistic effect, had better antifungal capacities than that of each individual component. Interestingly, the inhibition zone diameters of OCS/nSiO_2_ were approximately 4–5 mm and 7–9 mm larger than those of OCS and nSiO_2_, respectively [87]. Bio-synthetized MgO nanoparticles are produced using the native bacterial strain like *Bacillus* sp. The RNT3 strain was used to render CS-Mg nanocomposite. CS-Mg nanocomposite has demonstrated impressive antimicrobial activity against *Acidovorax oryzae* and *R. solani* and substantially inhibited development compared to the non-treated control [88]. In vivo assays with two plant hosts including tomatoes and peppers affected by *Fusarium* wilt and root rot diseases in which traditional chemical fungicides were used for comparative purposes displayed better antifungal activity of rGO-CuO NPs and a long-lasting impact at a very low concentration of 1 mg/mL. Interestingly, as CuO is a plant nutrient, the study of treated plants showed a positive impact on flowering, plant height and dry weight, as well as the aggregation of photosynthetic pigments [89].

## 4. Challenges

Food production and safety is a primary concern for all researchers to ensure that what we are consuming is a safe food free from contamination and an assurity of maintenance of the characteristic quality traits of the food. The primary problem of food is contamination by microbes affecting the plants at varying degree levels at seedling, rooting, flowering, and fruit stage, post-harvesting stage. Though we are practicing, routine intensive usage of chemical fertilizers, pesticides, inoculants, agrochemicals, and chemical sprayers during the postharvest period, but concerning the toxicity and accumulation of toxicants in food products causes major health allied problems in humans and also to the environment [3].

In this modern age of science, every decade is witnessing the advent of innovative scientific concepts and applications catering to the well-being of the human population and the environment. Likewise, this decade is comforted by the emergence of nanotechnology. This technology can resolve daunting problems persisting in all the sectors of physical, chemical, and biological sciences. In agriculture, the use of nanotechnology is indispensable owing to the reasons of superior properties.

Despite the usage of nanotechnology in agriculture, certain challenges have existed that need to be rectified or eliminated. In antifungal management, nanohybrid materials are being used. These consist of elements like silver, gold, copper, iron, graphene, silica; polymers like chitosan, PVC, PLGA, and other organic molecules that are incorporated to obtain composites or nano-hybrids. The nanohybrid production methods are cost-intensive since these involve the use of expensive chemicals, reagents, and physical energy. Thus, the produced nanohybrids may exhibit high effectivity against phytopathogens but on application under field conditions, these nanohybrids may exhibit off-target movements and may enter into the plant system or may get accumulated in the vegetative parts of the plants.

The effect of nanoparticles on crop plants must be understood before developing any kind of nano-formulations for antibacterial/fungicidal applications. Many researchers have suggested that nanoparticles can potentially harm plant growth and development. Dimpka et al. [92] reported that CuONPs can affect the root and shoot growth in wheat. Further, the chlorophyll content and enzyme activities peroxidase and catalase activities were reduced in CuONPs treated plants. Likewise, AgNPs at higher concentrations (10 mg/L) can have altered on the metabolism and cell defense mechanism in wheat [93]. TiO_2_ when exposed to *Oryza sativa* has shown to result in reduced biomass contents, changes in metabolite concentrations, and alterations in the respiration pathways [94]. Likewise, Carbon nanotubes reduced the length of root and shoot of a rice plant besides inducing the DNA damage [95].

Nanoparticles not only exert negative effects in plants but also may have altering effects on the soil microbial communities. All metallic/metal oxide nanoparticles reduce the microbial abundance at varying levels. *Pseudomonas putida* an important bacterium in nitrogen recycling was affected by carbon nanotubes and ZnONPs [96]. The same ZnONPs were also reported to have negatively affected the soil beneficial fungi also. Soil Microbial Biomass Carbon (SMBC) was observed to be drastically very low in fields where the soil microbes were exposed to AgNPs, NiNPs, CuONPs and carbon nanotubes [96]. The induction of reactive oxygen species (ROS) by Ag-SiO_2_ core-shell nanocomposites were responsible for radial hyphal growth inhibition of a few plant pathogenic fungi [97].

Another important challenge is that nanoparticles can trespass through food chains and accumulate at higher trophic levels. A report of Vittori Antisari et al. [98] documented that tomato plants exposed to engineered metal oxides resulted in increased concentrations of K and decreased in the Mg, P, and S contents in the fruits of tomato. When such fruit will be consumed by humans, their fate remains questionable. Therefore, the usage of nanoparticles remains a big and unresolved challenge for agricultural applications particularly involving their use in open field conditions.

## 5. Future Perspective

With the pros and cons of nanotechnology challenges; it is a time to explore the correct and beneficial usage of nanotechnology in agriculture management. To obtain the nanocomposite as a safe, clean, and eco-friendly agent for the management of fungal diseases in plants and postharvest period the following criteria can be accomplished. At present, chemical-based synthesis of nanocomposite is reported so far. However, the researchers are now endeavoring to fabricate nanocomposite through a green chemistry approach involving the utilization of agri-wastes such as banana or orange peels, wheat whiskers, straw, cotton or corn stalks, coconut or almond shells, corn silk, rice husks for the production of nanoscale carbon, silica, graphene, cellulose, and chitosan polymers. Further, tremendous opportunities of use and application of green synthesized inorganic metal/metal oxide nanoparticles can be identified in agriculture and particularly for the control and management of plant diseases caused by several fungal phytopathogens [99,100,101]. The biological synthesis protocols can improve the cost, time and energy requirements besides will help decrease the amounts of environment-corrosive chemicals required for the industrial production of nanomaterials and their composites through the most prevalent physical/chemical synthesis techniques [100]. The most striking benefit of the use of biodegradable polymers for development of nanocomposites by conjugation of metal/metal oxide nanoparticles is their ease of translocation within the plant tissues and the ability to exert in planta antifungal activity. Therefore, the use of biodegradable polymers must be encouraged in future research due to their biocompatible and eco-friendly characteristics.

Size and stability are two important factors for designing a novel nanocomposite. Producing a size-controlled nanocomposite will be a key success in antifungal management. Stability should be maintained till the end of the period. Before applying a nanocomposite material under field conditions; researchers should ensure the toxicity of the applied nanocomposite to the non-target organisms. Many nanocomposite materials are developed from toxic nanomaterials, for example, TiO_2_ is reported to produce colon cancer. Therefore, the derived nanocomposites can be toxic to plants, microbes, and the environment, and hence, a careful preparation of nanocomposite with minimal toxicity must be preceded.

The prepared nanocomposite must not exhibit undesirable effects in plants and fruits. In certain cases, the fruit ripening process may get delayed more than the expected period. Prompt application of nanocomposites as spray/emulsion can be encouraged during the post-harvest period (storage) (Figure 6). Novel nanomaterials (sensors, kits) should be developed to detect, quantify, and analyze the fungal pathogen during the post-harvest period. The toxicity of nano-composites depends on the concentration used. Compared to chemical-based pesticides/fungicides, the working concentrations of the nanocomposites are relatively very low. Another pressing challenge for the nano-products is hurdles faced in the marketing of these products possibly due to production cost, unclear technical benefits, public opinion, and legislative uncertainties. Compared to other sectors, the usage of nanotechnology in agriculture is marginal and needs attention.

## 6. Conclusions

A product that can attribute positive outcomes to our intended purpose must be welcome. Likewise, identification of the boons and banes of the nanocomposite smart materials as effective antagonistic agents to curb the fungal pathogens is critical. In this review, we have clearly emphasized the significance of nanocomposites in fungal disease management in a comprehensive approach. Post-harvest management of fruits by nanocomposites offers a successful tool to combat diseases and infections leading to produce loss through spoilage and decay.

Our findings have indicated that the control of toxigenic fungi and the detoxification of mycotoxins are not adequate for sustainable agricultural ergonomics. Therefore, novel treatment methods for improving the food safety and protection must be applied. Nanohybrid antifungals are thus, of primary importance for a synergistic approach to resolve diverse problems in the management of fungal pathogens causing agricultural/post-harvest diseases in the 21st century, with a focus on Green Nanotechnology, which is environmentally sustainable and provides a continuum for the plant, animal and human health. The nano-hybrid anti-fungals are anticipated to cater to the need of the growers, consumers as well as the environment activists through rapid, effective, and comparatively improved eco-safety attributes for controlling the yield and produce quality deterring potential of the fungal phytopathogens.

## Figures and Tables

**Figure 1 jof-07-00048-f001:**
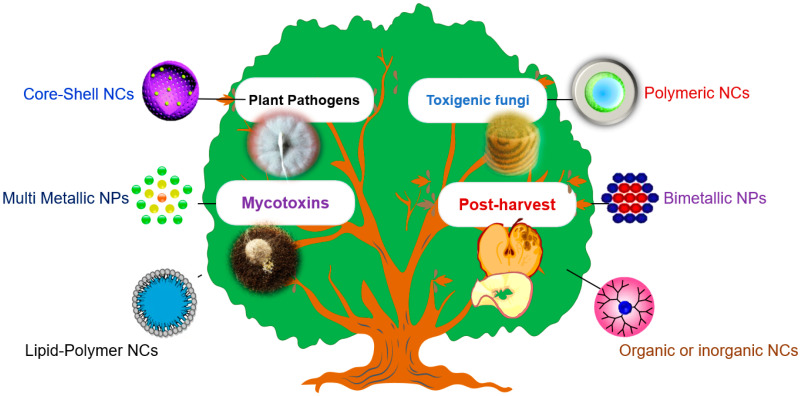
Nanohybrid antifungals for plant diseases control.

**Figure 2 jof-07-00048-f002:**
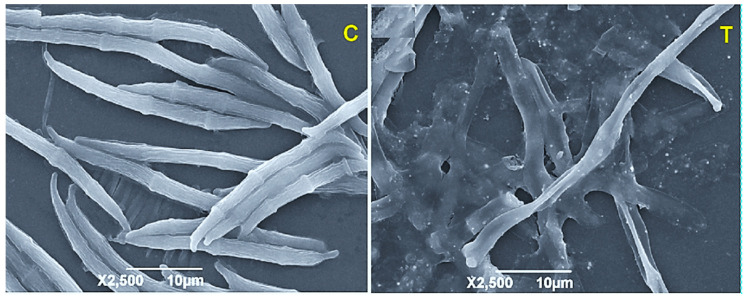
SEM images of *F. graminearum* spores incubated with sterile water (**C**) control and treated with GO-AgNPs nanocomposite (**T**). Images obtained with permission from Chen et al. [38]. (Cited from Chen et al., with permission from ACS).

**Figure 3 jof-07-00048-f003:**
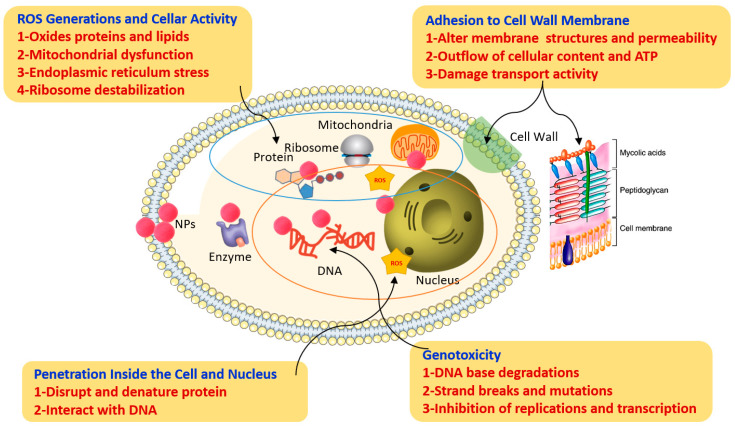
Antifungal activity mechanisms of hybrid nanomaterials.

**Figure 4 jof-07-00048-f004:**
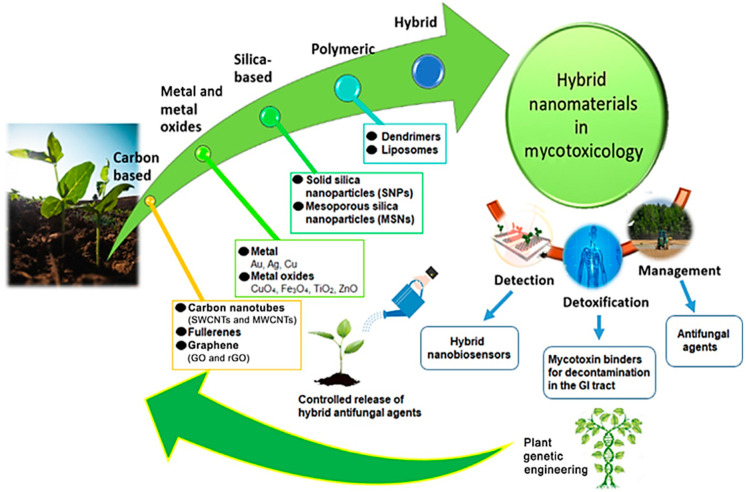
Hybrid-nanomaterial categories used in plant science: carbon nanotubes, including single-walled carbon nanotubes (SWCNTs) and multiwalled carbon nanotubes (MWCNTs), fullerenes); metallic and metal oxide NPs, silica-based nanostructures, and polymeric (dendrimers and liposomes) for the fabrication of biosensors for mycotoxin detection, detoxification of mycotoxin-contaminated food and feedstuffs through binders and management for sustainable control over fungal growth and mycotoxin contamination. (Cited from Thipe et al. with permission from Elsevier).

**Figure 5 jof-07-00048-f005:**
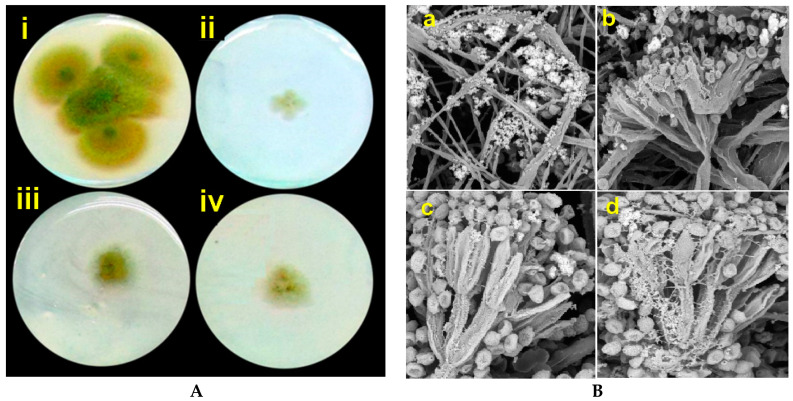
(**A**) Antifungal activity of *Ag-Chit NCs against*
*A. flavus* collected from feed samples. (i), control (without nanocomposite treatment), (ii), (iii), and (iv) fungal mat treated with 30, 60, and 90 milligrams of nanocomposites. All petri dishes treatment was incubated at 28 °C for 10 days. (**B**) Fungal mycelium of *P. expansum* treated with Ag-Chit NCs referred to the morphological changes in fungal hyphae, SEM images depicted markedly shriveled, crinkled cell walls, and flattened hyphae of the fungi (a), hyphal cell wall and vesicle damaged (b), irregular branching (a and b), and collapsed cell, formation of a layer of extruded material (d) Source (Abd-Elsalam KA. unpublished data).

**Figure 6 jof-07-00048-f006:**
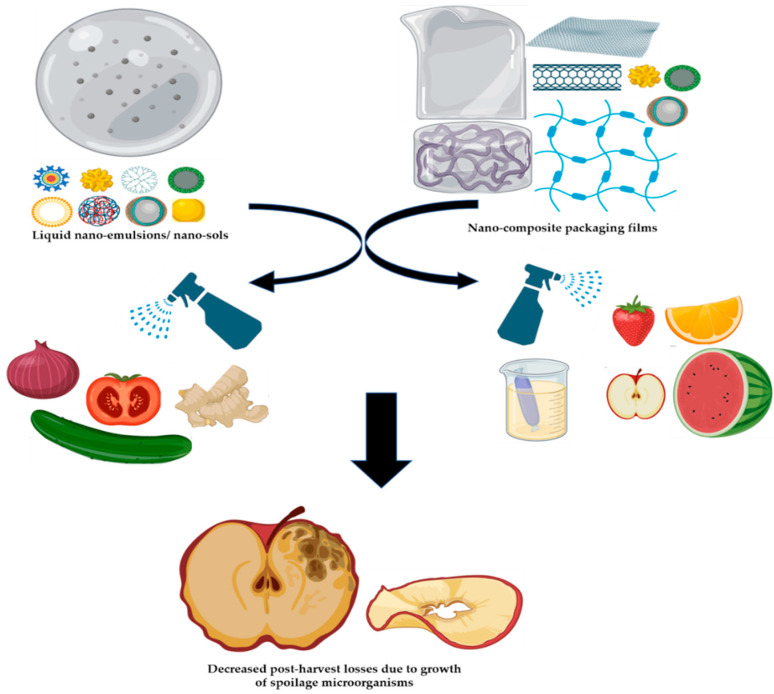
Application of nano-composite formulations to decrease post-harvest losses of horticultural produce.

**Table 1 jof-07-00048-t001:** Nano-composite formulation and their application for curbing the plant fungal pathogens.

Type of Nanocomposite Applied	Method of Synthesis	Effective Working Concentration	Application Method	Pathogen Studied	Remarks	References
**Inorganic-inorganic composites**						
AgNPs-titanate nanotubes	Photo-assisted functionalization of AgNPs on hydrothermal micro-wave-assisted synthesis of titanate nanotubes	30 mg/30 mL	In vitro study performed in PDB supplemented with 30 mg photo-activated AgNP-titanate nanotube composite	*Botrytis cinerea*	Fungal conidial death due to ROS damage	[40]
Fe_3_O_4_/ZnO/AgBr	Microwave-assisted synthesis	1:8 weight ratio nanocomposite	In vitro spore broth incubation study performed in a cylindrical Pyrex reactor	*Fusarium graminearum, Fusarium oxysporum*	Complete inactivation of test fungi within one hour of incubation with the nanocomposite	[41]
Bimetallic (Au/Ag) NPs with metal oxide NPs(ZnO NPs)	Physical mixture technique	50:10 μg/mL	In vitro poison food study involving the addition of NP suspension in SDB	*Aspergillus flavus/A. fumigatus*	-Augmented inhibition of fungal growth by bimetallic and metal oxide NPs	[90]
ZnO:Mg(OH)_2_ composite	Hydrothermal/co-precipitation technique	5 to 0.002 mg mL^−1^	In vitro study involving DMSO dissolved NPs supplemented in PDB	*Colletotrichum gloeosporioides*	-Addition of MgO diminished the antifungal potential of ZnO NPs	[91]
**Inorganic-carbon composites**						
CuO NPs functionalized graphene-like carbon composite	Green synthesis using *Adhatoda vasica* leaf extract and 0.01 M CuSO_4_	5:4 ratio proportion of leaf extract: CuSO_4_	In vitro agar well diffusion assay on PDA media	*Aspergillus niger, Candida albicans*	growth inhibition due to the disruption of the cell membranes	[42]
GO-AgNPs	Interfacial electrostatic self-assembly synthesis	9.37 µg/mL-MIC value	In vitro assay using growth media and detached wheat leaf bioassay	*Fusarium graminearum*	Improved anti-fungal efficiency [>3-fold for AgNPs and >7-fold over pure GO] through two mechanisms (physical injury and ROS mediated chemical injury)	[38]
**Inorganic-organic composites**						
ZnO NPs/CS-Zn-CuNPs	Wet chemical method	0 to 90 µg mL^−1^	In vitro study involved supplementation of nanocomposite in PDA media	*Alternaria alternata, B. cinerea, R. solani*	-Highest mycelial inhibition by chitosan mixed Zn-Cu nanocomposite	[48]
Cu-/Zn-chitosan and bimetallic nanocomposites	Wet chemical synthesis	30, 60, and 100 μg mL^−1^	-In vitro study using agar based media -In vivo seed priming assay for damping-off disease in cotton cultivar Giza 92 seedlings	*Rhizoctonia solani*	-highest hyphal inhibition at 100 μg mL^−1^ -Augmented effect of bimetallic NC along with biocontrol fungus (*Trichoderma*)to suppress disease in vivo	[49]
Clay-chitosan nanocomposite	Anion-exchange technique	5 to 60 μg mL^−1^	-In vitro study using PDA -In vivo assay in mature fruits of *Citrus sinensis* (L. Osbeck) cv. Valencia	*Penicillium digitatum*	-Complete inhibition of fungal hyphae in different weight ratios of clay/chitosan nanocomposite (1:0.5, 1:1, 1:2) (conc.20 μg mL^−1^)	[47]

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
