# Peer review of "Nanohybrid Antifungals for Control of Plant Diseases: Current Status and Future Perspectives"

_jof, 2021, doi:10.3390/jof7010048_

Round 1

Reviewer 1 Report

Fungal diseases cause signifi cant economic agricultural losses around the world and their control have been limited to chemical fungicide in an irrational manner. Nanotechnologies highly support the implementation of innovative strategies for the control of plant diseases, even if to understand the possible benefi ts of employing nanoparticles to agriculture it is necessary to analyze the penetration and transport of nanoformulations in plants and animals. However, the present manuscript reviews the development of antifungal nanocomposites, in a perspective of the microbial resistance occurrence mitigation and the containment of crop yield losses due to fungal pathogens.

I found the manuscript complete enough, even if some issues have been risen; some of them are listed as follows:

An English language revision is recommended, since frequently the use of the grammar and the syntax results quite weird.

The Author's line contains many errors in names, affiliation attribution numbers and punctuation/spaces

Abstract

Line 29: "Food and its products are of paramount importance to humans;" this sentence is needless. Delete.

Line 30: "microbes" it would be more correct to use "pathogens" or "microbial pathogens"

Line 31: what did Authors mean with "the increasing human population and deleterious attack on vegetation by microbes are worsening the research fraternity in a melancholy way"? It has no sense.

Line 40: "antifungal" change with "fungal"

Keywords are too generic to be applied as real keywords to the manuscript. Others more specific of the topic discussed should be chosen.

Introduction:

The introduction needs to be enriched with the bibliography from Perez-Alvarez et al. (Plant Fungal Disease Management Using Nanobiotechnology as a Tool. In book: Advances and Applications Through Fungal Nanobiotechnology), that in 2016 presented a review on the use of nanotechnologies against fungal plant diseases. I found that Authors should introduce such bibliography explaining why their current review is different and/or more updated. This is mandatory, otherwise the presen manuscript could appear as a remastered copy.

Line 74: "applicationsparticular instance" please fix.

Line 87: "Alumino" change with "Alumina" or "Aluminum"

Line 87: citation [12] sould be implemented with bibliography from Spadola et al. 2020 ("Thiosemicarbazone nano-formulation for the control of Aspergillus flavus" doi: 10.1007/s11356-020-08532-7.")

Line 94: "cheaper, reliable and environment-friendly product(s)" my suggestion is to not ensure the "environmental friendship" of such compounds, since many of them proved to induce cytotoxic and genotoxic side effect on animals and humans, needing therefore an accurate evaluation of their negative effect prior an extended use in agriculture.

Line 97: "the entire crop field will get destroyed" is too dramatic; change with "the crop yield might result heavily compromised"

Line 106: "research fraternity" that's incorrect. Please use "research community"

Line 116: "5 phytoplasma μL-1" did Authors mean "5 phytoplasma cells" or CFU? Please clarify.

Line 125: "While Wang et al." syntax error

Line 248: "[22]" again, the work of Spadola et al. 2020 ("Thiosemicarbazone nano-formulation for the control of Aspergillus flavus" doi: 10.1007/s11356-020-08532-7.") is missing: since it describes the nanoformulation of antifungals/antiaflatoxigenic/ antisclerotigen compounds in polyvinyl alcohol (PVA), it perfectly fit with the discussion.

Chapter 3.3 (Nanohybrid antifungals for controltoxigenic fungi and mycotoxins degrdtions) should be placed before the chapter about the "Postharvest management of nanocomposite against pathogenic fungi" (3.2), since the interference with the secondary fungal metabolism mainly relies on in field applications, at the pre-harvest stage. Additionally, due to the relationship between mycotoxins biosynthesis and fungal cell redox balance, it would be more appropriate to make it follow the previous chapter on ROS-dependent machanism of action. This section should be improved with a more detailed discussion about the effectiveness of nanoparticles in modulating/preventig mycotoxins accumulation on crops, along with their capacity of reducing the formation of dispersal structures (such as not only spores but also sclerotia), that, in turn, can  limit the spread of mycotoxigenic species in the environment.

Line 441: "degrdtions" please fix.

Figure 4: "A. falvus" change with "A. flavus"

Author Response

Reviewer 2:

I had a great opportunity to asses review manuscript entitled: „Nanohybrid antifungals for management of plant diseases: Current status and future prospective” which is considered for publication in Journal of Fungi.  I read carful whole manuscript and it is well written and presents new and valid use of nano-antifungals. I think that manuscript could be improved by some changes listed below:

Comment 1: Authors should add markings of nanocomposite and fungus on photos what is which it will clarify this figure.

Answer: The markings have been incorporated in the images to clarify the contents.

Comment 2: Figure 2 I thinks that yellow blocs (for example with ROS Genrations) with text should be more connected with central part of this figure.

Answer:

Comment 3: Figure 3 should be much much bigger. This change will outline the meaning of this figure.

Answer: The image size has been enhanced as desired.

Comment 4: Figure 4. It is little problematic because markings of photos. All markings is small letters if petri photos is complementary to SEM photos I suggest to make photos in pairs petri dish and SEM. If not then I suggest to make markings from A-H instead a-c and again a-c.

Answer: The markings have been changed as indicated.

Reviewer 2 Report

Dear Authors,

I had a great opportunity to asses review manuscript entitled: „Nanohybrid antifungals for management of plant diseases: Current status and future prospective” which is considered for publication in Journal of Fungi.  I read carful whole manuscript and it is well written and presents new and valid use of nano-antifungals. I think that manuscript could be improved by some changes listed below:

  1. Authors should add markings of nacomposite and fungus on photos what is which it will clarify this figure.
  2. Figure 2 I thinks that yellow blocs (for example with ROS Genrations) with text should be more connected with central part of this figure.
  3. Figure 3 should be much much bigger. This change will outline the meaning of this figure
  4. Figure 4. It is little problematic because markings of photos. All markings is small letters if petri photos is complementary to SEM photos I suggest to make photos in pairs petri dish and SEM. If not then I suggest to make markings from A-H instead a-c and agan a-c.

Sincerely,

Author Response

Reviewer 1:

Comments and Suggestions for Authors

Fungal diseases cause significant economic agricultural losses around the world and their control have been limited to chemical fungicide in an irrational manner. Nanotechnologies highly support the implementation of innovative strategies for the control of plant diseases, even if to understand the possible benefits of employing nanoparticles to agriculture it is necessary to analyze the penetration and transport of nanoformulations in plants and animals. However, the present manuscript reviews the development of antifungal nanocomposites, in a perspective of the microbial resistance occurrence mitigation and the containment of crop yield losses due to fungal pathogens. I found the manuscript complete enough, even if some issues have been risen; some of them are listed as follows:

Comment 1: An English language revision is recommended, since frequently the use of the grammar and the syntax results quite weird.

Answer: The manuscript has been thoroughly revised to omit the errors in use of English language.

Comment 2: The Author's line contains many errors in names, affiliation attribution numbers and punctuation/spaces

Answer: The names and affiliation attribution numbers have been rechecked. The punctuation and spaces errors have been removed.

Comment 3: Abstract Line 29: "Food and its products are of paramount importance to humans;" this sentence is needless. Delete.

Reply: The sentence has been deleted as desired.

Comment 4: Line 30: "microbes" it would be more correct to use "pathogens" or "microbial pathogens"

Answer: The term microbe has been replaced with ‘pathogens’ or ‘microbial pathogens’.

Comment 5: Line 31: what did Authors mean with "the increasing human population and deleterious attack on vegetation by microbes are worsening the research fraternity in a melancholy way"? It has no sense.

Answer: The sentence has been modified as desired.

Comment 6: Line 40: "antifungal" change with "fungal".

Answer: The desired change has been incorporated.

Comment 7: Keywords are too generic to be applied as real keywords to the manuscript. Others more specific of the topic discussed should be chosen.

Answer: The keywords have been replaced with more specific ones.

Comment 8: Introduction: The introduction needs to be enriched with the bibliography from Perez-Alvarez et al. (Plant Fungal Disease Management Using Nanobiotechnology as a Tool. In book: Advances and Applications Through Fungal Nanobiotechnology), that in 2016 presented a review on the use of nanotechnologies against fungal plant diseases. I found that Authors should introduce such bibliography explaining why their current review is different and/or more updated. This is mandatory, otherwise the manuscript could appear as a remastered copy.

Answer: We thank the reviewer to put forth this viewpoint. The manuscript by Alvarez et al (2016) details out the role of different types of nanoparticles (169-172), fungal disease scenario (172-177), mycosynthesis of silver, gold, and cadmium sulphide nanoparticles (177-180), and inorganic-based nanofungicide (181-182). Only a single paragraph summarizing the ‘polymer-based/ hybrid nanofungicides’ has been included. However, in this review the focus is on the development of anti-fungal nano-hybrids which include a variety of nanocomposites derived from inorganic-inorganic, inorganic-organic and organic-organic matrices-nanomaterial filler agents. Therefore, this manuscript is substantially different from contents discussed in the Alvarez et al manuscript and should not be considered a remastered copy of Alvarez et al (2016) book chapter.     

Comment 9: Line 74: "applicationsparticular instance" please fix.

Answer: The desired change has been incorporated.

Comment 10: Line 87: "Alumino" change with "Alumina" or "Aluminum"

Answer: Alumino-silicates are primarily soil clay minerals comprised of oxides of aluminium and silicon and certain counter-cations. The hyphen has been included in between two words as a correction. Same word has also been used in Alvarez et al (2016) manuscript at page number 171.

Comment 10: Line 87: citation [12] should be implemented with bibliography from Spadola et al. 2020 ("Thiosemicarbazone nano-formulation for the control of Aspergillus flavus" doi: 10.1007/s11356-020-08532-7.")

Answer: The indicated research paper will be incorporated where the nanomaterial based  

Comment 11: Line 94: "cheaper, reliable and environment-friendly product(s)" my suggestion is to not ensure the "environmental friendship" of such compounds, since many of them proved to induce cytotoxic and genotoxic side effect on animals and humans, needing therefore an accurate evaluation of their negative effect prior an extended use in agriculture.

Answer: The word environment-friendly has been removed as suggested by the reviewer.

Comment 12: Line 97: "the entire crop field will get destroyed" is too dramatic; change with "the crop yield might result heavily compromised"

Answer: The phrase has been replaced with  "the crop yield might result heavily compromised".

Comment 13: Line 106: "research fraternity" that's incorrect. Please use "research community"

Answer: The word ‘fraternity’ has been replaced with ‘community’ as desired.

Comment 14: Line 116: "5 phytoplasma μL-1" did Authors mean "5 phytoplasma cells" or CFU? Please clarify.

Answer: Yes, it means sensitive identification of Candidatus even at occurrence of 5 cells per microliter. The desired changes have been incorporated to improve the clarity.

Comment 15: Line 125: "While Wang et al." syntax error.

Answer: The modification has been incorporated to address this issue.

Comment 16: Line 248: "[22]" again, the work of Spadola et al. 2020 ("Thiosemicarbazone nano-formulation for the control of Aspergillus flavus" doi: 10.1007/s11356-020-08532-7.") is missing: since it describes the nanoformulation of antifungals/antiaflatoxigenic/ antisclerotigen compounds in polyvinyl alcohol (PVA), it perfectly fit with the discussion.

Answer: The work of Spadola et al. (2020) has been cited at the place indicated in the manuscript.

Comment 17: Chapter 3.3 (Nanohybrid antifungals for control toxigenic fungi and mycotoxins degrdtions) should be placed before the chapter about the "Postharvest management of nanocomposite against pathogenic fungi" (3.2), since the interference with the secondary fungal metabolism mainly relies on in field applications, at the pre-harvest stage. Additionally, due to the relationship between mycotoxins biosynthesis and fungal cell redox balance, it would be more appropriate to make it follow the previous chapter on ROS-dependent mechanism of action. This section should be improved with a more detailed discussion about the effectiveness of nanoparticles in modulating/preventing mycotoxins accumulation on crops, along with their capacity of reducing the formation of dispersal structures (such as not only spores but also sclerotia), that, in turn, can limit the spread of mycotoxigenic species in the environment.

Answer: The suggested additions have been performed in the manuscript. The additions in the section have been incorporated as indicated by the reviewer.

Comment 18: Line 441: "degrdtions" please fix.

Answer: The mis-spelled word has been corrected.

Comment 19: Figure 4: "A. falvus" change with "A. flavus"

Answer: The mistyped word has been corrected.

Reviewer 3 Report

The manuscript entitled Nanohybrid antifungals for management of plant 3 diseases: Current status and future prospective to JOF Journal.

The concept of the manuscript is novel, fits and suitable to publish in JOF Journal. This manuscript is generally well written and clearly presented however still need to address many comments and thus require substantial major revision before its acceptance.

  • Provide a nice graphical abstract representing the overview of the MS with key highlights. Give full form of abbreviation in the abstract as well as in whole manuscript.
  • Title should be modifying instead of 'management' it should be 'control' and 'perspective' to 'perspectives'.
  • In the introduction section, write the novelty of the work and the problem statement clearly. Give details of usefulness of NPs synthesis methods and more substantial discussion on green synthesis methods is expected refer and cite important review articles Colloids and Surfaces B: Biointerfaces 170, 20-35, 2018, Environmental Science and Pollution Research 25 (11), 10164-10183, 2018.
  • Section titles should be modifying and describe in details. For section 1 give more details by citing and refereeing recent review of literature.
  • Figure 4 give source of information. Section 3 refer and cite Environmental Science and Pollution Research 25 (11), 10392-10406, 2018. Still more discussion of toxicity of NPs for the phytopathogens control by adding a new section is expected.
  • More details about the practical applications and future research perspectives and challenges is expected Moreover what authors think about this should be mentioned.
  • The conclusion of the study is not discussed with the specific output obtained from the study, it could be modified with precise outcomes with a take home message.
  • English and grammar mistakes are present. The author should check the manuscript by native English Speaker to improve the quality of the manuscript.

Author Response

Reviewer 3:

Comments and Suggestions for Authors

The manuscript entitled Nanohybrid antifungals for management of plant 3 diseases: Current status and future prospective to JOF Journal. The concept of the manuscript is novel, fits and suitable to publish in JOF Journal. This manuscript is generally well written and clearly presented however still need to address many comments and thus require substantial major revision before its acceptance.

Comment 1: Provide a nice graphical abstract representing the overview of the MS with key highlights. Give full form of abbreviation in the abstract as well as in whole manuscript.

Answer: The graphical abstract has been prepared. Also, the abbreviations in the abstract and whole manuscript are being updated.

Comment 2: Title should be modifying instead of 'management' it should be 'control' and 'perspective' to 'perspectives'.

Answer: The title has been modified as suggested.

Comment 3: In the introduction section, write the novelty of the work and the problem statement clearly. Give details of usefulness of NPs synthesis methods and more substantial discussion on green synthesis methods is expected refer and cite important review articles Colloids and Surfaces B: Biointerfaces 170, 20-35, 2018, Environmental Science and Pollution Research 25 (11), 10164-10183, 2018.

Answer: The introduction is improved by addition of relevant content on problem statement and novelty of the work. The general detail regarding the NPs synthesis methods have been incorporated. The indicated references have been incorporated in the relevant section. 

Comment 4: Section titles should be modifying and describe in details. For section 1 give more details by citing and refereeing recent review of literature.

Answer: The introduction section has been improved as indicated. The section titles and the contents have been reshuffled as indicated.

Comment 5: Figure 4 give source of information. Section 3 refer and cite Environmental Science and Pollution Research 25 (11), 10392-10406, 2018. Still more discussion of toxicity of NPs for the phytopathogens control by adding a new section is expected.

Answer: The source for Figure 4 has been added in the manuscript. The antifungal properties of the nanocomposites and NPs have been further elaborated in section 3.1 and 3.2. The indicated reference has been cited in the future perspective section.   

Comment 6: More details about the practical applications and future research perspectives and challenges is expected. Moreover what authors think about this should be mentioned.

Answer: The details have been updated with contents pertaining to the our viewpoint on the use of nanohybrids as potential antifungals.

Comment 7: The conclusion of the study is not discussed with the specific output obtained from the study, it could be modified with precise outcomes with a take home message.

Answer: The conclusion has been modified to incorporate the precise outcome.

Comment 8: English and grammar mistakes are present. The author should check the manuscript by native English Speaker to improve the quality of the manuscript.

Answer: The English and grammar mistakes have been removed.

Round 2

Reviewer 3 Report

Authors have substantially revised the manuscript according to comments. The present form is acceptable and to publish in JOF.